# Incorporating Domain Knowledge and Structure-Based Descriptors for Machine Learning: A Case Study of Pd-Catalyzed Sonogashira Reactions

**DOI:** 10.3390/molecules28124730

**Published:** 2023-06-13

**Authors:** Kalok Chan, Long Thanh Ta, Yong Huang, Haibin Su, Zhenyang Lin

**Affiliations:** Department of Chemistry, The Hong Kong University of Science and Technology, Clear Water Bay, Kowloon, Hong Kong SAR, China; klchanbl@connect.ust.hk (K.C.); long.ta@ust.hk (L.T.T.)

**Keywords:** activation energy, homogeneous catalysis, ligand effects, machine learning, phosphine ligands

## Abstract

Machine learning has revolutionized information processing for large datasets across various fields. However, its limited interpretability poses a significant challenge when applied to chemistry. In this study, we developed a set of simple molecular representations to capture the structural information of ligands in palladium-catalyzed Sonogashira coupling reactions of aryl bromides. Drawing inspiration from human understanding of catalytic cycles, we used a graph neural network to extract structural details of the phosphine ligand, a major contributor to the overall activation energy. We combined these simple molecular representations with an electronic descriptor of aryl bromide as inputs for a fully connected neural network unit. The results allowed us to predict rate constants and gain mechanistic insights into the rate-limiting oxidative addition process using a relatively small dataset. This study highlights the importance of incorporating domain knowledge in machine learning and presents an alternative approach to data analysis.

## 1. Introduction

Machine learning has become an increasingly popular tool for data analysis in various fields [1]. Its strong pattern recognition and predictive ability have prompted chemists to investigate its applications in several subfields of chemistry [2,3]. In physical chemistry, the prediction of pKa through Lewis structures has been achieved with the aid of a large database [4,5]. In computational chemistry, machine learning force fields have been developed to integrate the performance of ab initio methods and the efficiency of classical force fields [6,7,8,9,10,11]. In retrosynthetic chemistry, deep learning, a subfield of machine learning, has been applied to achieve automated retrosynthesis planning based on a large pool of published reactions [12,13]. In material chemistry, machine learning has aided in predicting material properties, such as those of OLED [14] and crystals [15].

With the evolution of high-throughput experimentation and easily available online databases, chemists now have access to large datasets of high quality to perform predictions of reaction yield, selectivity, and condition optimization. Jenson, Barzilay, and co-workers [16] trained a graph convolutional neural network (based on patent literature) for product prediction with over 400 k data points. Applying a highly advanced deep learning architecture based on BERT, a recent robust language model pretrained on over 10 k reactions, Schwaller and co-workers [17] implemented reaction SMILES as inputs and predicted the yields of Suzuki–Miyaura reactions with good performance. Schwaller and co-workers [18] also employed an unsupervised transformer encoder model in conjunction with SMILES to generate products-to-reactants atom-mapping for reactions such as Diels–Alder reactions and epoxidations using 2.8 M reactions.

Studies involving machine learning are often based on a large amount of data as mentioned above, including both experimental and/or DFT-calculated data, which are, however, difficult to obtain and require high expenditure and effort for data collection and processing. In reality, data acquired from chemical research, especially in the area of organic reaction studies, are often very limited in size. Thus, when applying machine learning to study chemical reactions with a limited set of data, chemists start to intuitively employ chemically meaningful descriptors in order to derive accurate predictions. For instance, Doyle, Dreher, and co-workers [19] managed to predict the yields of Ni-catalyzed Suzuki–Miyaura cross-coupling of benzaldehyde-derived acetals with aryl boroxines with high accuracy by using generated and empirical spatial descriptors, such as buried volume and Tolman cone angle, using around 4000 data points. In another study, Doyle and co-workers [20] used computed atomic and empirical molecular properties (e.g., electrostatic charge, NMR shift) as well as binary categorical identifiers to predict reaction yields for the deoxyfluorination of a broad range of alcohols with sulfonyl fluorides using random forest based on 640 reactions. Yada, Sato, and co-workers [21] predicted the reaction yields for a tungsten-catalyzed epoxidation of alkene with hydrogen peroxide by coupling DFT-calculated descriptors with the least absolute shrinkage and selection operator methods, a regression analysis method commonly used in machine learning, with 3800 reactions. Denmark and co-workers [22] used feed-forward neural networks coupled with electronic descriptors and novel 3D steric descriptors generated from DFT calculations to predict the selectivity of phosphoric acid-catalyzed thiol addition reactions using around 1000 data points.

Encouraged by the above-mentioned successful applications of machine learning with limited data sizes in chemistry, we attempt to go one step further and study chemical reactions of an even smaller dataset by simply utilizing structural descriptors and traditional electronic descriptors with a specially defined machine learning architecture without the need for DFT calculations. It is worth noting that useful descriptors have been established to study Diels–Alder reactions. For example, DFT-calculated descriptors, such as global electrophilicity power difference (Δω) [23], and empirical descriptors, such as Taft’s polar and steric substituent constants [24], have been employed to describe their activation energies.

Here, we choose to study a palladium-catalyzed Sonogashira cross-coupling reaction using trialkyl phosphine ligands, reported in 2008 by Plenio and co-workers [25] and shown in Figure 1. Using high-throughput experiments, the authors studied and determined the rate constants of 340 reactions consisting of 17 alkyl phosphine ligands and 20 different meta- and para-substituted aryl bromide substrates. The 340 data points (rate constants) formed a relatively small dataset, which we used here to showcase our deep learning study.

We generated descriptors based only on the topological structures of the phosphine ligands. For the electronic bias of the aryl bromide substrates, we chose a widely used and chemically meaningful descriptor—the Hammett constant—to capture the chemical reactivity of the aryl bromide reagents involved. Using the rather small dataset, we demonstrate that the new machine learning architecture we built performs very well and is able to derive well-known/accepted knowledge regarding the rate-limiting steps of Pd-catalyzed Sonogashira cross-coupling reactions, suggesting the usefulness of the protocol presented in this work.

## 2. Results

### 2.1. Performance of Our Ligand Descriptors

The averaged predicted rate constants of the 10 selected optimal sets of trainable parameters for the training, validation, and testing datasets were plotted against the experimental values in Figure 1. To directly compare with the experimental values, we chose to plot the rate constant instead of the corresponding converted free energies. We achieved a high R^2^ of 0.94 on the training dataset, 0.87 on the validation dataset, and 0.84 on the testing dataset. The good performance on the testing dataset demonstrated our proposed machine learning model’s high predictive ability.

### 2.2. Comparison to Other Ligand Descriptors

To demonstrate the advantage of using the tree-like structural descriptors (for the phosphine ligands) over others, we examined four other types of common descriptors, two of which are based on DFT-calculated structures: buried volumes and cone angles. The buried volumes and cone angles for the 17 phosphines were calculated using the scheme reported in Kraken, a phosphine database with comprehensive physicochemical descriptor information [26]. The other two types of descriptors are based on one-hot encodings and multiple molecular fingerprints [27]. We conducted machine learning studies using these 4 types of descriptors with two machine learning methods: The first method is to modify the architecture we discussed in Section 4.2.1 by replacing the GNN layer and its output (the vertex values) with new descriptors. This modification was necessary because all four descriptors cannot be adequately represented graphically and thus are not directly compatible with GNN. The second method is to employ only the final layer of the architecture we built, i.e., a restricted linear regression. More specifically, we fit the linear regression model with positive coefficients and zero intercept against the rate constant k. This restriction was applied so that a fair comparison can be made between the results using different descriptors. Since we aim to examine different ligand descriptors, the aryl bromide Hammett constant descriptor remains unchanged. Table 1 summarizes the learning performances using these different types of descriptors and compares them against that using the tree-like structural descriptor.

The performance comparison presented in Table 1 clearly indicates that, within the rather small dataset studied in this work, the tree-like structural descriptors outperformed the other types of descriptors by a significant margin. We also attempted machine learning by combining both buried volume and cone angle as input descriptors. The performance remained inferior to our tree-like structural descriptors. Both buried volume and cone angle, although excellent descriptors reflecting ligand steric effect, did not perform well with the dataset studied in this work. Although one-hot encoding and multiple fingerprint features (MFF) performed better than buried volume and cone angle, they were much worse than the tree-like structural descriptors.

### 2.3. Cross-Validation

To evaluate the performance of our model, we conducted k-fold cross-validation as follows: We partitioned the dataset into 5 equal-sized datasets (S_1_, S_2_, S_3_, S_4_, S_5_), where 4 subsets (S_1_–S_4_) were utilized as training or validation sets and one subset (S_5_) was used exclusively as the testing set and was not involved in any training processes. Therefore, the testing set remained constant across all four cross-validation sets. To create the validation set, we selected one subset from S_1_–S_4_. This resulted in a total of 4 distinct cross-validation sets, namely, sets 1, 2, 3, and 4, which utilized S_1_, S_2_, S_3_, and S_4_ as their validation set, respectively. It is worth noting that the performance of set 1 is identical to the performance displayed in Figure 1. Table 2 shows that the performance of both the validation set and testing sets is greater than 0.75 and 0.8, respectively, for all cross-validation sets. Additionally, the average performance of the validation sets and testing sets are 0.84 and 0.87, respectively, which confirms that our models possess a high level of generalizability. Based on the results of cross-validation, we believed that the overfitting problem associated with our model should not be significant. In our model, we incorporated regularization techniques such as Leaky ReLU and L1 and L2-regularizers, leading to minimization of overfitting.

## 3. Discussion

Our results indicate that the tree-like structural descriptors exhibited superior performance compared to other existing descriptors. Here, we demonstrate how our protocol can provide mechanistic details that are consistent with the current understanding of cross-coupling reactions. As previously mentioned, we define ΔG^‡^ as the sum of ΔG^‡^(L) and ΔG^‡^(L-S) to segregate the effect of purely ligand component and the combined effect of ligand and substrate. This machine learning architecture allows us to obtain the fitted values ΔG^‡^(L) and ΔG^‡^(L-S) separately and analyze their corresponding trends.

### 3.1. ΔG^‡^(L)

As shown in Figure 2, the fitted ΔG^‡^(L) values were sorted and plotted against Ligands 1–17. A boxplot method was used to display the results predicted by the 10 optimal sets of trainable parameters selected with the criteria mentioned in Section 4.2.2.

Figure 2 reveals that the least steric ligand, L1 (tri-n-butylphosphine, ^n^Bu_3_), has the highest ΔG^‡^(L), while the bulkiest ligand, L16 (di(1-adamantyl)benzylphosphine, (1-Ad)_2_PBn), has the lowest ΔG^‡^(L). The results are consistent with our general understanding that bulky phosphines promote the OA process.

According to a kinetic study by JF Hartwig and F Barrios-Landeros [28], in the palladium-catalyzed Sonogashira coupling of aryl bromides using an exceptionally bulky ligand, Q-phos, the reaction rate only depends on ligand concentration. Therefore, they concluded that the rate-determining step is ligand dissociation. In the dataset we used, the rate-determining step is oxidative addition. The obtained ΔG^‡^(L) values offer a reconciliation of these two observations. Bulkiness of the ligand negatively correlates with ΔG^‡^(L). Therefore, for an exceptionally bulky ligand such as Q-phos, the oxidative addition transition state likely lies even lower in energy than the ligand dissociation transition state.

### 3.2. ΔG^‡^(L-S)

Unlike ΔG^‡^(L), which depends only on ligand, ΔG^‡^(L-S) is affected by both ligand and aryl bromide substrate. Thus, we plotted ΔG^‡^(L-S) against ligand for each substrate and against substrate for each ligand (see Appendix A for the details). Examination of these plots revealed that all the plots of ΔG^‡^(L-S) against ligands showed a similar trend, as did all the plots of ΔG^‡^(L-S) against substrates.

To understand and discuss how ligands affect ΔG^‡^(L-S), we plotted ΔG^‡^(L-S) against ligands for 1-bromo-4-nitrobenzene substrate as an example, which exhibited the widest range of fluctuation, as shown in Figure 3. From this figure, it appeared that the effect of ligand on ΔG^‡^(L-S) differed from that on ΔG^‡^(L), in which ΔG^‡^(L) showed a strong correlation with the steric property of ligands while ΔG^‡^(L-S) did not. Generally, bulky ligands lead to higher ΔG^‡^(L-S) because of increasing steric crowdedness at the Pd center in the transition state. Therefore, the change of ΔG^‡^(L-S) with respect to ligands reflected the combined effect of the phosphine ligand, including both electronic and steric. In the case of alkylphosphines, the electronic difference among ligands was relatively small and the effect of ligand on ΔG^‡^(L-S) was mostly steric.

Next, we present the plot of ΔG^‡^(L-S) against aryl bromides for Ligand 7 (L7) in Figure 4. From this plot, the Hammett constant of the substituent is negatively correlated with ΔG^‡^(L-S). Aryl bromides with a more electron-withdrawing substituent (i.e., a more positive Hammett constant) had lower ΔG^‡^(L-S), while those with a more electron-donating substituent (i.e., a more negative Hammett constant) had higher ΔG^‡^(L-S). This trend aligned with the general understanding of oxidative addition [29].

### 3.3. ΔG^‡^

The previous two subsections discussed the correlations of ΔG^‡^(L) and ΔG^‡^(L-S) with L (ligand) and S (substrate). To understand the general trend of these reactions, we have to examine the total activation free energy, ΔG^‡^. Thus, we plotted ΔG^‡^ against ligand for 1-bromo-4-nitrobenzene in Figure 5. The plot reveals that bulkier phosphine ligands have smaller ΔG^‡^. A previous experimental study [30] showed that pCy_3_ as a ligand in Sonogashira coupling tends to be modest, while CyP*^t^*Bu_2_ was a better ligand. This experimental observation is consistent with the trend predicted in Figure 5 that CyP*^t^*Bu_2_ (L12) gives smaller ΔG^‡^ than pCy_3_ (L3).

While the general trend related to the steric factor exists, it is clearly not the only factor influencing the reactivity trend. For example, EtPAd_2_ (L11) is bulkier than P*^t^*Bu_2_Bn (L15), but the former gives a larger ΔG^‡^, thus is less reactive. The same reverse trends are also observed for Cy_2_pAd (L8) vs. Cy_2_P*^t^*Bu (L9) and CyP^t^Bu_2_ (L12) vs. ^i^Pr_2_P*^t^*Bu (L10). In other words, the cooperative nature between ligand and substrate may lead to a scenario where we cannot simply employ a single factor to explain experimental observation. The study by Plenio and coworkers [30] also demonstrates that in addition to the steric factor, electronic factor of phosphine ligands are also important for their reactivity.

Moreover, Figure 5 shows that ΔG^‡^(L-S) is generally greater than ΔG^‡^(L), implying that ΔG^‡^(L-S) has a greater contribution to the overall predicted ΔG^‡^. Comparing the plots in Figure 3 and Figure 4, we observe that the change of ΔG^‡^(L-S) with ligands is narrower (Figure 3) than that with substrates (Figure 4), suggesting that the effect of ligands on ΔG^‡^(L-S) is not as significant as that of the substrates. This conclusion agrees with the experimental observation by Hartwig and coworkers that the reaction rate of oxidative addition of bromoarenes is positively related to bromobenzene concentration and only weakly dependent on ligand concentration [31].

## 4. Methodology

As stated in the Introduction, our goal is to apply machine learning to a relatively small dataset of rate constants for Pd-catalyzed Sonogashira cross-coupling reactions. To achieve this goal, incorporating pertinent chemical knowledge is crucial in defining chemically meaningful descriptors and constructing a robust machine learning framework. Therefore, let us briefly discuss the current understanding of the reaction mechanisms first. For palladium-catalyzed cross-coupling reactions, it is widely accepted that the general catalytic cycle involves three fundamental steps [32,33]: oxidative addition of Pd(0) to an aryl halide to form a Ar-Pd(II)-X complex, transmetallation between the nucleophile and the Ar-Pd(II)-X complex, and reductive elimination to regenerate Pd(0) and produce the final coupling product. It is also widely understood that in most of these cross-coupling reactions, oxidative addition is often rate-limiting [34,35]. The Sonogashira cross-coupling reaction studied here is also an example of this.

Extensive and detailed kinetic studies [32,33,34,35] on the oxidative addition (OA) of aryl halides to phosphine Pd(0) complexes have concluded that the specific OA mechanisms mainly depend on both the steric properties of the phosphine ligands and the electronic properties of the aryl halides. The possible OA mechanisms include both associative and dissociative pathways (Figure 2), and their variants. The literature reported 340 Sonogashira cross-coupling reactions that involve a wide range of phosphines and aryl bromides. Thus, it is not reasonable to assume that a single OA pathway can account for the rate-determining oxidative addition step. It is expected that the various mechanistic scenarios presented in Figure 2 are all possible.

### 4.1. Descriptors

#### 4.1.1. Descriptors for Aryl Bromides

Based on the above mechanistic analysis, the rate-determining step (OA) likely involves the cleavage of the C-X bond on the Pd(0) metal center. The more electron-deficient the C-X bond is, the easier the OA becomes, i.e., the lower the activation barrier ΔG^‡^. In the reactions shown in Figure 1, various aryl bromides with different substituents were used as substrates. The extent of electron deficiency of the C-Br bond in an aryl bromide largely depends on the identity and position of the substituent on the aryl ring. Thus, Hammett σ constants [36] for substituent groups were chosen as electronic descriptors for aryl bromides because they are widely accepted as electronic parameters for substituted arenes and used in quantitative structure-activity relationship (QSAR) studies. It should be pointed out here that Hammett σ constants were also used to correlate with Sonogashira cross-coupling reactivity of aryl bromides in the work by Plenio and co-workers [25], from which the measured rate constants dataset is drawn for this study. The 20 electrophile substrates used in the Pd-catalyzed Sonogashira cross-coupling reactions were aryl bromides substituted with an electron-withdrawing or electron-donating group at either *meta*- or *para*-position. Since steric effects of substituents at *meta*- and *para*-positions are generally considered less significant compared to *ortho*-positions, and considering that most substituents in the dataset were not bulky, the bulkiest substituent, -*^t^*Bu, located at *para*-position and far from the reaction center, should also be negligible sterically. Thus, steric factors likely did not contribute significantly to the reactivity of the underlying electrophiles and were not considered. Each aryl bromide was represented by a vector with 2 elements (σ*_meta_*, σ*_para_*), which contains information relevant to the Hammett constant of the substituent and its location, *meta* or *para*. Since there was only one substituent, either at *meta* or *para* position, on each aryl bromide, the unsubstituted aromatic position was assigned as 0 (the Hammett constant for a hydrogen substituent). For instance, a *meta*-bromobenzonitrile was represented as (σ_m_^CN^, 0), while a *para*-bromobenzonitrile was represented as (0, σ_p_^CN^).

#### 4.1.2. Descriptors for Phosphine Ligands

In palladium-catalyzed cross-coupling reactions, the structure of the phosphine ligand is crucial. The steric topology of phosphine ligands is frequently a primary consideration in ligand design, which contributes significantly to the OA activation barrier ΔG^‡^. Hence, to retain structural information without using any DFT data, we employed graphical representations to describe the 3D molecular structure of a phosphine. We found it convenient to use a tree-like representation to describe a phosphine ligand. The most important atom of a phosphine molecule, phosphorus, was considered the “root”, or reference point of the tree-like layout, while carbon atoms that radiate from the root are called “nodes”. The nodes were organized in a layer-by-layer format, in which nodes with the same smallest number of bonds away from the root were considered to be in the same layer: The nodes that are connected to the phosphorus atom via a single bond (i.e., carbon atoms directly linked to the root) were categorized in the first layer. The nodes that are two bonds away were classified in the second layer, and so on.

Using this tree-like framework, all ligands were mapped as shown in Figure 6a. The 1st layer nodes were prioritized by their substituents according to the Cahn–Ingold–Prelog rules, which are commonly used for assigning stereochemistry. Similarly, the nodes in other layers were assigned using the same procedure, constituting a unique representation (alignment graph) for each phosphine ligand molecule.

The alignment graph for each phosphine ligand molecule was converted to an input graph consisting of node feature data and edge feature data. A Boolean (1 or 0) representation was utilized to denote the presence of a node on the graph, where 1 indicates existence and 0 indicates non-existence. The edges were characterized by the bond order, where 1 denoted a single bond, 1.5 referred to a C-C bond in an aromatic ring, and 2 indicated a double bond. More information on how the model handles such data will be elaborated in the subsequent section.

### 4.2. Machine Learning

#### 4.2.1. Machine Learning Architecture

Incorporating domain knowledge of the aforementioned mechanisms that dictate the rate-determining oxidative addition step is essential for designing our machine learning model. The practice of incorporating domain knowledge in machine learning has been reported in the literature. For instance, Amal and co-workers [37] proposed integrating bandgap-related physics equations into machine learning to study photocatalysis, and Riley and co-workers [38] applied rule-based chemistry restrictions, e.g., forbidding the formation of additional bonds among non-adjacent atoms within the same ring when using deep reinforcement learning models for drug discovery.

Before being processed by machine learning, the rate constants were first converted into activation energies using the Eyring equation, which resembles the Arrhenius equation (Equation (1)):(1)$$k=\frac{k_{B}T}{h}e^{-\frac{E_{a}}{RT}}$$
where *E_a_* is the activation energy, *k_B_* is Boltzmann’s constant, *h* is Planck’s constant, and *R* is the gas constant. As a result, the output from the machine learning model would be the prediction of the activation free energies (ΔG^‡^) that were derived from the experimentally measured rate constants.

Considering the various mechanistic scenarios shown in Figure 2 and the well-established notion that the phosphine ligand is the most important factor in palladium-catalyzed cross-coupling reactions [39,40], we proposed that ΔG^‡^ can be reasonably estimated by a linear combination of two parts: (1) ΔG^‡^(L), dependent purely on ligand, reflecting the steric effect of phosphine, and (2) ΔG^‡^(L-S), dependent on both ligand and substrate, reflecting their cooperative nature in the OA step. We will integrate this proposal into our machine learning framework to improve prediction accuracy, given the relatively limited dataset. The most convenient and reasonable way to parse the previously defined tree-like structural descriptors of the phosphine ligand is using a graph-based method, a graph neural network (GNN). Thus, ligand-dependent values (h_L_) were generated by parsing the ligand structural descriptors to a GNN layer to produce vertex values, followed by a single layer of fully connected neural network (FCNN). The ligand substrate-dependent values (h_L-S_) were generated by parsing both the vertex values obtained from the GNN layer for the ligand and the aryl bromide descriptors (Hammett constants) to another single layer of FCNN, as shown in Figure 6b. The activation energy ΔG^‡^ (output) would be the sum of ΔG^‡^(L), derived from the scaled h_L_, and ΔG^‡^(L-S), derived from the scaled h_L-S_.

#### 4.2.2. Machine Learning Model Training Process

Following the framework illustrated above (Figure 6b), neural networks were built with the packages TensorFlow 2.7 [41] and Deep Graph Library [42] (DGL, for GNN layer) in the Python language. Adaptive Moment Estimation (Adam) optimizer was used with Mean Absolute Error (MAE) as the loss function. Hyperparameters were adjusted using the Optuna package [43], including the learning rate, the message dimension, and the number of iterations for the GNN layer. Further details related to hyperparameters, regularizers, weight restrictions, and early termination aspects are provided in the Section 5.

The dataset was randomly divided into three subsets: a training dataset (60%, 204 data points), a validation dataset (20%, 68 data points), and a testing dataset (20%, 68 data points). By applying the setup described above, we trained the neural networks by batch gradient descent with a batch size of 10 samples from the training dataset for 10,000 epochs. To select optimal values of trainable parameters, the coefficient of determination of R^2^ > 0.8 and the mean absolute error (MAE) of MAE < 1.5 from the validation set were used as the thresholds. Only those neural networks delivering results meeting both criteria were selected. The selection of these thresholds was empirical, considering and balancing the number of required trials and the performance of the selected models subsequent to the model selection process. Final predictions were calculated by averaging the predicted results from applying each of the 10 optimal sets of trainable parameters with the largest R^2^ on the validation set. Cross-validation was carried out for further validation, which is discussed in more detail in a later section.

## 5. Computational Details

The general formulae for the structure of the GNN layer in a message-passing neural network (MPNN) framework are displayed as Equations (2) and (3). As an example, we focus on one of the atoms—atom *a*_1_ in a molecular structural input. Let *a_x_* be the label of any neighboring atom of *a*_1_. For the first iteration, i.e., time step *t* = 0, $h_{a_{1}}^{0}$ is the node feature (also referred to as the initial hidden state) of *a*_1_, $h_{a_{x}}^{0}$ is the node feature of *a_x_*, and $e_{a_{1}a_{x}}$ is the edge (bond) feature of *a*_1_ and *a_x_*. Then the summation of all the message functions (*M_t_*) results of each *a_x_* gives $m_{a_{1}}^{1}$, the “message” of *a*_1_ at *t* = 1, as shown in Equation (2):(2)$$m_{a_{1}}^{t+1}=\sum_{a_{x}\in Neighbor}M_{t}\left(h_{a_{1}}^{t},h_{a_{x}}^{t},e_{a_{1}a_{x}}\right)$$

This message $m_{a_{1}}^{1}$, together with the current node feature $h_{a_{1}}^{0}$, will next be used as inputs for the update function, *U_t_*, to generate a new node feature $h_{a_{1}}^{1}$, as shown in Equation (3):(3)$$h_{a_{1}}^{t+1}=U_{t}\left(h_{a_{1}}^{t},m_{a_{1}}^{t+1}\right)$$

For the next iteration, the same process will be carried out, in which the previous node feature inputs (old hidden states) $h_{a_{1}}^{0},h_{a_{x}}^{0}$ will be replaced by new hidden states $h_{a_{1}}^{1},h_{a_{x}}^{1}$ respectively. This process will end if a predefined number of iterations is reached. The final hidden state will be the input of the following layer.

As mentioned in the Section 4.2.2, some of the hyperparameters were chosen using the Optuna package. Learning rates were chosen from 0.0005, 0.001, 0.002, and 0.005. The detailed workflow of the model and the corresponding hyperparameters were described as follows:Phosphine ligand descriptors were passed through a GNN layer with 3–5 iterations in which each iteration had the same weight without bias. The GNN layer was constructed in the framework of a message-passing neural network (MPNN) using Leaky ReLU activation with a = 0.01, size 2–4 for message function, and size 1 for update function. The graphical output was concatenated according to the order of an aligned graph. The generated vector was referred to as GNN Output States.Aryl bromide descriptors and GNN Output States were concatenated and passed through a fully connected neural network (FCNN) layer using Sigmoid activation, size 1, L2-regularizer, and no bias. The generated vector was referred to as Hidden States 1.GNN Output States were passed through an FCNN layer using Sigmoid activation, size 1, L2-regularizer, with non-negative weights and no bias. The generated vector was referred to as Hidden States 2.Hidden States 1 and Hidden States 2 were passed through an FCNN layer using linear activation, size 1, with non-negative weights and no bias, generating predicted activation energies.Predicted activation energies were converted to reaction rate constants using the Eyring equation.

Within a maximum of 10,000 epochs in the training process, the epochs were distributed between two stages: In the first stage, in order to increase the training efficiency, only steps 1–4 were performed, i.e., the outputs were the predicted activation energies, while the corresponding ground truths were the activation energies calculated from the experimentally measured rate constants. The first stage ended when the cutoff (<5.0, <6.0 or <7.0 kcal/mol in loss) was reached. The remaining epochs were considered in the second stage. In the second stage, in order to achieve finer optimization of the prediction, steps 1–5 were performed, i.e., the outputs were the predicted reaction rate constants, and the corresponding ground truths were the experimental reaction rate constants. Early termination of the training process would be activated if the loss fluctuates to a certain degree. This fluctuation was calculated by the mean absolute value of the last 10 batch losses. If 80% of the last 10 batch loss changes show different directions when compared with their previous loss changes, i.e., one was an increase while another was a decrease, and the mean absolute value of such loss changes exceeds 0.2, the training process would be considered highly fluctuated and thus rejected. On the other hand, if the mean absolute value of the loss changes < 0.0001, the training process would be considered as converged and terminated early.

## 6. Conclusions

In summary, we developed a simple yet highly effective tree-like representation for phosphine ligands that can be utilized to generate input graphs for GNNs. By combining this ligand descriptor with a well-established electronic descriptor of aryl bromides (Hammett constant) and integrating human knowledge of mechanistic details into our machine learning model, we were able to extract ligand- and substrate-specific activation energies (ΔG^‡^(L) and ΔG^‡^(L-S)), enabling us to investigate the impact of ligands and substrates on the rate-limiting step. Our machine learning protocol confirmed previously established chemical principles regarding cross-coupling reactions. In terms of practicality, our approach only required the tree-like representations (generated easily from the chemical structure) and straightforward empirical electronic descriptors for the substrates (Hammett σ constant of its substituent) as input. It was rewarding to develop a knowledge-driven machine learning model for chemical reaction prediction using a relatively small dataset. Our model not only provided accurate predictions but also shed light on the underlying scientific questions.

## Data Availability

The code and data used to produce the reported results can be found online at: https://github.com/klchan4207/PdSonoML.

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
