# Peer review of "Incorporating Domain Knowledge and Structure-Based Descriptors for Machine Learning: A Case Study of Pd-Catalyzed Sonogashira Reactions"

_molecules, 2023, doi:10.3390/molecules28124730_

Round 1

Reviewer 1 Report

In this work, the authors developed a machine learning based method to predict rate constants of chemical reactions of interest. Below are my comments:

1. line 121-124: references are needed to show previous studies

2. line 124-127: references are needed to show previous studies

3. line 129: The authors need to give reasons why steric factors are not considered.

4. line 177-179: The authors need to give more details (mathematical formula) to show how the experimental/predicted rate constants were converted to activation free energies.

5. line 210: The authors need to provide more explanation to clarify the cutoff used for R2 and MAE.

6. Uncertainty estimate is missing for reported R2 in line 222 and data reported in Table 1. I suggest the authors perform bootsrtapping analysis to get error bars for these data (similar to the way the authors did for cross validation).

Some typos and grammar errors are detected

Author Response

Dear reviewer,

Please find the responses to your kind comments.

Thank you very much!

Haibin Su and Zhenyang Lin

Reviewer 2 Report

In their manuscript, "Domain Knowledge and Structure-Based Descriptors for Machine Learning: A Case Study of Pd-Catalyzed Sonogashira Reactions", Chan et al. present a meticulously-crafted study that employs a graph-based neural network to predict kinetics of Sonogashira reactions. The design of descriptors is reasonable and the model surpasses other simpler descriptors. This performance is expected given the capability of GNN models to encapsulate more topological features. I support this manuscript's publication in Molecules, although there are a few minor issues that need addressing:

1) The MAE values are mentioned throughout the manuscript, but without units.

2) Due to accessibility issues, I wasn't able to view the SI. I am not certain if the authors have included parity plots for the activation energies directly. Considering the final rate values were transformed using the Arrhenius equation, the error distribution for k could be significantly more amplified than that for G. An analysis of how such non-linear conversion impacts the final error would be enlightening.

3) Following my previous comment, creating a parity plot for G (or perhaps a log-log scale for k) could prove beneficial to visualize the data point distribution.

4) If my understanding is correct, the descriptor size for the ligand is much larger than that of the reactant. Could such design risk downplaying the impact of the reactant on G(L-S)? Did the authors consider a model featuring a graph representation of the reactant, or perhaps one with weighted contributions?

5) Similar to point 4, given that the descriptors for G(L) and G(L-S) are quite similar, there might be some "error cancellation" affecting the final prediction results, where the errors in predicting individual G(L) and G(L-S) values partially nullify each other in the combined output G. Could the authors verify this possibility?

Author Response

Dear Reviewer,

Please find the responses to your kind comments.

Thank you very much!

Haibin Su and Zhenyang Lin

Reviewer 3 Report

In this manuscript, the authors suggested a machine learning model for predicting the activation free energies of Sonogashira cross-coupling reactions using the topological structures of phosphine ligands as the descriptors. As a result, they constructed GNN model using tree-like representations for the phosphine ligands. Though this study may be a more appropriate model than conventional models that are constructed using standard structural parameters, I think that this study contains unfair parts and lacks the analysis of the obtained models. So, I recommend that this manuscript should be majorly revised for the publication. Followings are other points that have to be revised:

1. The authors use not conventional simple structural parameters but the topological parameters of phosphine ligands focusing on the oxidative addition reactions. I think it is more appropriate to be based on conventional experimental findings. However, in this case, the authors should issue a general guideline for reflecting such experimental findings to the models. Otherwise, it requires the reconstruction of the model and data collection for machine learning for each case, though it takes too long to get useful results earlier than to perform the experiments.

2. The characteristics of the models obtained in the machine learning are not explained. It is necessary to discuss whether the characteristics are useful to design new ligands. If not, it is hard to avoid the possibility that the model occasionally provides accurate predictions.

3. This study focuses only on predicting the activation free energies determining the rate constants. However, the interest of experimentalists is not limited to the rate constants. The authors should mention about the important properties that will be predicted by the related studies.

4. This manuscript contains some typos and strange English phrases.

It needs English check by a native speaker.

Author Response

(The authors gave the same response as above.)
